# Influence of intrauterine factors on birth weight and on child linear growth in rural Ethiopia: A prospective cohort study

**Meselech Roro**[1,2]*, **Wakgari Deressa**[2], **Bernt Lindtjørn**[1]

**1** Centre for International Health, University of Bergen, Bergen, Norway, **2** School of Public Health, College of Health Sciences, Addis Ababa University, Addis Ababa, Ethiopia

* meselua@yahoo.com

## Abstract

### Introduction

Little is known about the influence of intrauterine fetal factors on childhood growth in low-income countries. The objective of this study was to examine the influence of intrauterine fetal growth on child linear growth in rural Ethiopia.

### Methods

We conducted a prospective community-based cohort study from July 2016 to October 2018. All pregnant women with gestational age of 24 weeks or below living in 13 kebeles, in central Ethiopia were enrolled. The fetuses were followed from pregnancy up to 11–24 months after birth. We measured biparietal diameter, head circumference, femoral length, and abdominal circumference at 26, 30 and 36 weeks of pregnancy. At birth, we measured infant weight. At 11–24 months of age, z-scores of length- for- age, and weight-for-length were calculated. A multilevel, mixed-effect, linear regression model was used to examine the influence of fetal, newborn, maternal, household factors and residence area on child linear growth.

### Results

We included 554 children. The prevalence rate of stunting was 54.3% and that of wasting was 10.6%. Fetal biparietal diameter, head circumference, and abdominal circumference, were significantly associated with birth weight. Femoral length z-score in early pregnancy, gestational age at delivery and child age were significantly associated with length-for-age z-score. Family size was significantly associated with length-for-age z-score. Family size and maternal height were associated with weight-for-height z-score. There was a large variation in length-for-age z-score (Intra cluster correlation, or ρ (rho) = 0.30) and weight-for-length z-score (ρ = 0.22) than of birth weight of new-born (ρ = 0.11) in kebeles indicating heterogeneity in clusters for length-for-age z-score and weight-for-length z-score than birth weight.

**Data Availability Statement:** All data files are available from the Open Science Framework (OSF). Here is the link to the data: osf.io/t9n8q.

**Funding:** This study was funded by University of Bergen in Norway and the Research Council of Norway (project number: 220554). The funders had no role in study design, data collection and analysis, decision to publish, or preparation of the manuscript. The author(s) received no specific funding for this work.

**Competing interests:** The authors have declared that no competing interests exist.

## Conclusions

Child linear growth was influenced by fetal growth, duration of pregnancy, maternal height, and family size. Environmental factors that are associated with the area of residence play a bigger role for linear growth than for birth weight.

## Introduction

One of the most important development phases of a child is the first 1,000 days of life, which is the time from conception up to two years of age [1, 2]. Early childhood developments happen in physical, social, emotional, and cognitive domains. If there is any incident that happens during pregnancy or in the child's first years of life, it can affect growth and lead to childhood undernutrition [2–4]. Childhood undernutrition is a major, common public health problem that increases the risk of mortality among children in developing countries [5]. Stunting affects 164.8 million (22.7%) children globally. Most of the stunted children, 148 million, live in low- and middle-income countries (LMICs). Additionally, 52 million (8%) of children worldwide exhibit signs of wasting [5, 6].

Studies of the timing and pattern of growth faltering in the first two years of life suggest that the growth rate of children is adequate in the first few months of life and growth faltering often starts after three months of life [7, 8]. In LMICs, the length/height for age z-scores (LAZ) declines soon after birth to a lowest point of -1.75 to < 2 z-scores by 24 months of age. This often results in lack of subsequent catch-up growth up to the age of five years [9].

Birth cohort studies suggest that growth in the first 1,000 days of life can be viewed as a continuum between the fetal period and infancy and early childhood period [9, 10]. Some factors that contribute to growth faltering in low resource settings include inappropriate breastfeeding, lack of adequate quality and amount of complementary foods, insufficient infant and young child feeding practices, infections, and other environmental exposures [2, 9, 11]. However, little is known about the influence of intrauterine fetal factors on childhood growth in low-income countries. It is thus important to examine the effect of intrauterine fetal growth on early childhood growth. The aim of this study was to examine the influence of intrauterine fetal growth on LAZ and weight-for-length z-score (WLZ) in early childhood using a prospective cohort study conducted starting from 24 weeks of gestation up to 11–24 months of age in rural Ethiopia.

## Methods

### Study design

We conducted a prospective community-based cohort study from July 2016 to October 2018. All pregnant women with gestational age of 24 weeks or below living in 13 kebeles of Adami Tullu, Oromia, Ethiopia were enrolled. We followed the fetus, using portable obstetric ultrasound, starting from the date of enrolment throughout the period of pregnancy and then until 11–24 months of age, when weight and length were measured.

The influence of fetal biometry on child linear growth was examined in this study. The fetal biometry in millimeters measured by ultrasound included were biparietal diameter (BPD), head circumference (HC), femoral length (FL), and abdominal circumference (AC). The other independent variables we included were child sex, gestational age at birth, maternal age, maternal education, maternal occupation, parity, maternal height, wealth status, and family

size. The outcome measurements for child linear growth were length–for–age z-scores (LAZ) and weight-for-length z-scores (WLZ). The other intermediate outcome measurement that was evaluated for its effect on linear child growth was birth weight.

We identified and adapted independent variables that influence linear growth in children aged 11–24 months of age from previous studies [2, 3, 12, 13]. In the current study, the socio-economic conditions were family size and household wealth status. The maternal conditions were age, height, parity, education and occupation. The fetal and child factors include fetal factors (HC, BPD, AC and FL) and birth characteristics (gestational age at birth and child sex).

## Study setting

This study recruited pregnant women living in 13 kebeles (lowest administrative level) of the Adami Tullu district, Oromia Regional State, Ethiopia, as previously reported [14]. At baseline, trained data collectors interviewed all women of reproductive age regarding socio-demography, household conditions and pregnancy status.

The projected population size of the district for 2015 was 177,390 people [15]. Local residents primarily depend on farming, livestock rearing, and to a lesser extent, fishing in Lake Zeway for their subsistence.

## Participants

The study population and recruitment of study participants have been described in detail in previous publication [14]. Briefly, recruitment of the study participants was conducted from July 2016 to June 2017. All women of reproductive age (15–49 years) permanently residing in the study area were used as a source population for the current fetal and postnatal growth study.

Trained data collectors conducted house-to-house visits and informed eligible women about the study objectives and procedures. Eligible participants self-reported their pregnancy status or answered a modified checklist based on 2010 criteria endorsed by the World Health Organization (WHO) [16]. Pregnancy was confirmed by ultrasound, and those with a gestational age at or before 24 weeks of gestation who gave written informed consent (thumbprint or signature) were enrolled. We followed the BPD, HC, AC and FL parameters of the fetus using ultrasound through the pregnancy. At delivery, we measured birth weight within in 72 hours after delivery. Subsequently their weight and length were measured after birth by the time they were 11–24 months of age.

We instructed the participants to attend three prenatal visits that were prescheduled from 26 weeks to 36 weeks of gestation at the closest health post or established stations in each kebele for this study. The data collectors reminded the study participants a day prior to their scheduled visits.

## Variables

Only singleton fetuses under 24 weeks of gestation estimated by ultrasound at enrolment were included in the analyses. Twins and fetuses with congenital anomalies were excluded. The outcome measurements for child linear growth were length-for-age z-score (LAZ) and weight-for-length z-score (WLZ), measured at 11–24 months of the child age and birth weight. Fetal biometry measured by ultrasound, sex of child, gestational age at birth (measured in weeks), maternal age, maternal education, maternal occupation, parity, maternal height, household wealth status, family size and child age were the predictor variables in this study. A household wealth status was constructed by using principal component analysis as described previously [14]. The number of household family members were used for family size.

All variables included in the model were considered as continuous except for occupation of the mother and sex of the child.

## Data collection procedures

To collect data we used interview questionnaires, anthropometric measurements, and ultrasound examinations. We used questionnaires that were structured and pre-tested to collect data on maternal age, socioeconomic status, education, occupation, and parity at enrolment. We developed the questionnaires in English and then translated into the local language, *Afan Oromo*. We trained thirteen nurses for data collection. They were trained on how to take maternal, new-born, and child anthropometric measurements. We assigned them to each health post or kebele station established for the study. Fig 1 summarizes the data collection timeline and variables.

## Ultrasound measurements

We calculated gestational age at enrolment based on the ultrasound measurement of biometric parameters using the Hadlock et al multiple parameter formula. The measurements were BPD, HC, AC, and FL [17, 18]. These were taken for pregnancies more than 13 weeks and 4 days. Assessments were performed at 24–28 weeks, 29–33 weeks, and 34–38 weeks of gestation. Even though sex of the fetus was identified during ultrasound assessment, the mothers were not informed about the sex of the fetus either by the researcher or by research assistant. For all ultrasound examinations and measurements, we used Sonosite M-Turbo diagnostic full-colored ultrasound machine (FUJIFILM SonoSite, Inc., USA). The primary author was trained on obstetric ultrasound. During training, the quality and accuracy of measurements were validated by a senior obstetrician at Addis Ababa University. After training, the primary author conducted the ultrasounds using standard techniques [19] and Hadlock's criteria [17, 18].

## Anthropometric measurements

Trained nurses recorded the anthropometric measurements. They measured height of each pregnant woman at enrolment using a standard wooden board. Trained nurses recorded the anthropometric measurements. They measured height of each pregnant woman at enrolment

**Fig 1. Data collection timeline of the study.**

using a standard wooden board. Before starting the study, they were given two days of training on anthropometry measurement techniques. We checked inter-and intra-technical measurement errors, and gave repeated training until the measurements reached the recommended cut-off points [20]. Within 72 hours of birth, the nurses measured birthweight, and recorded sex of the new-born. The weights and lengths of the children were recorded once at their age between 11–24 months. Length was measured in supine position (recumbent length) using a standard wooden board. Weight was measured to the nearest 100 gram using a digital scale [21]. The data collectors for the child anthropometry in this study were the nurses who were trained and their measurement techniques standardized in a study that was being carried out at the same time in under five children in the same area [22]. The intra and inter technical errors of measurements (TEM) were within the acceptable cut-off points [23]. They were given additional one-day refreshment training before initiation of data collection. Before taking weight measurements, the scales were checked and calibrated. Birth weight was measured within 72 hours of delivery to minimize variations related with physiological changes that occur after 72 hours.

Age of the children was calculated in months from the day on which anthropometric measurements were taken and the recorded date of birth.

## Sample size

The sample size was determined based on birth weight. The aim was to have an adequate sample to detect a birth weight difference of 110 grams among rich and poor based on wealth status and with previous estimates of standard deviation of birth weight being about 500g [24, 25]. The calculated sample size in each group was 325 (power of 80%). The overall sample size became 716 considering a non-response rate of 5% and a loss to follow-up of 5%.

## Statistical analysis

Trained clerks entered the data from the paper-based standardized questionnaires and forms using SPSS version 24 (SPSS Inc., Chicago, IL, USA). Data were cleaned using SPSS statistical software. We used Stata software version 15 (Stata Corp. College Station, TX, USA) for analysis. Descriptive statistics were calculated using the mean and standard deviation for continuous variables and percentages for categorical variables. In addition, scatter plot, boxplot and Inter Quartile Ranges were used for descriptive analysis. Principal component analysis was used to construct a relative wealth index [26]. It was computed using fourteen household assets. The detail was presented elsewhere [14].

The three rounds of fetal biometry at 24–28 weeks, 29–33 weeks and 34–38 weeks of gestation and birthweight were considered separately. Nutritional indices (standard deviations) and z-scores for length-for-age, weight-for-age, and weight-for-length were calculated using child growth standards [27]. Outcomes of less than -2 z-scores for length/height-for-age, weight-for-age, and weight-for-length/height were defined as stunting, wasting and underweight, respectively.

To measure the association of intrauterine growth on birth outcomes and growth among children 11 to 24 months of age, we used a multilevel, linear regression analysis model. This was to account for clustering due to repeated measures of HC,BPD,AC, and FL of the individual child at different times [28]. The other reason we used multilevel model was to account for determining factors of child linear growth (LAZ, WLZ) at household and community level that are nested in households and kebeles [29].

We first checked for distributions of the independent and outcome variables before fitting the model. We had planned to include the birth length in the model. Unfortunately, the length

measurements at birth was not correctly taken, and were excluded from the final analysis. Fitting of the multilevel model was done after checking for the presence of clustering. Initially, we fitted a null, single-level (standard) regression model. Then we ran a null, multilevel model with the random individual effect. The multilevel, linear regression analysis model was fitted for each exposure variables separately with the outcome variable and the final model was fitted for all exposure variables together.

The calculated test statistics for model fit of outcome measures was strong (P<0.001). To estimate the unadjusted and adjusted coefficient (β) with a 95% confidence interval (CI), we fitted a multilevel model to account for clustering.

The potential predictor variables considered for birth weight, LAZ and WLZ in the model included household, maternal, and child characteristics. The household characteristics were household wealth and family size. The maternal characteristics included age, height, education, occupation, and parity. While the child characteristics were HC, BPD, AC, FL, gestational age at delivery and sex. We also included all predictive variables as potential risk factors for birth-weight outcome. Finally, we conducted multilevel, mixed-effect, regression analyses for different times and reported the adjusted beta. Kebele was used as group identity and Random-effect parameters were estimated for it. In the statistical analyses, sex and gestational age at birth were factor considered as potential effect modifiers while other factors were considered as potential confounders.

Intra cluster correlation (ICC (ρ (rho)) was estimated as the ratio of the between kebele variance to the total variance in birth weight, length-for-age z-score and for weight-for-length z-score. ICC was calculated after a mixed-effects linear model was fitted. An ICC close to one for a variable shows homogeneity in a kebele. An ICC close to zero for a variable reflects that the variable is randomly distributed among kebeles [30].

## Ethical approval

We obtained ethical approval for this study in June 2015 from Addis Ababa University College of Health Sciences Institutional Review Board (ref.; 005/15/SPH) and from the Regional Committee for Medical and Health Research Ethics, Western Norway (ref: 2013/986/REK Vest). We also obtained written permission from the Oromia Regional Health Bureau and respective local authorities. All women who volunteered to participate in the study gave informed written consent and we obtained parental consent for their children.

## Results

### Characteristics of study participants

We screened 1,054 women of reproductive age of whom seven hundred twenty-seven pregnant women fulfilled the inclusion criteria while 327 did not meet the inclusion criteria. From the 727 pregnant women we enrolled, 23 of the pregnant mothers out-migrated before the date of delivery. From the 704 remaining study participants, 29 were excluded because of fetal or maternal causes. From those excluded, 12 of them were multiple pregnancies and 11 of the pregnancies terminated without any interference. Four pregnancies, which were referred to hospital after identifying congenital abnormalities, were terminated while two were intrauterine fetal death. Subsequently, we followed 675 fetuses throughout the pregnancy up to birth (Fig 2).

At birth, we took birth weight for 610 new-borns within 72 hours of delivery. However, birth weight for 65 new-borns were not taken. Three of them were stillbirths. Four of the new-borns weight measurements were not taken in 72 hours of birth and were thus excluded. In addition, 58 of the pregnant mothers left the study area at the time of delivery.

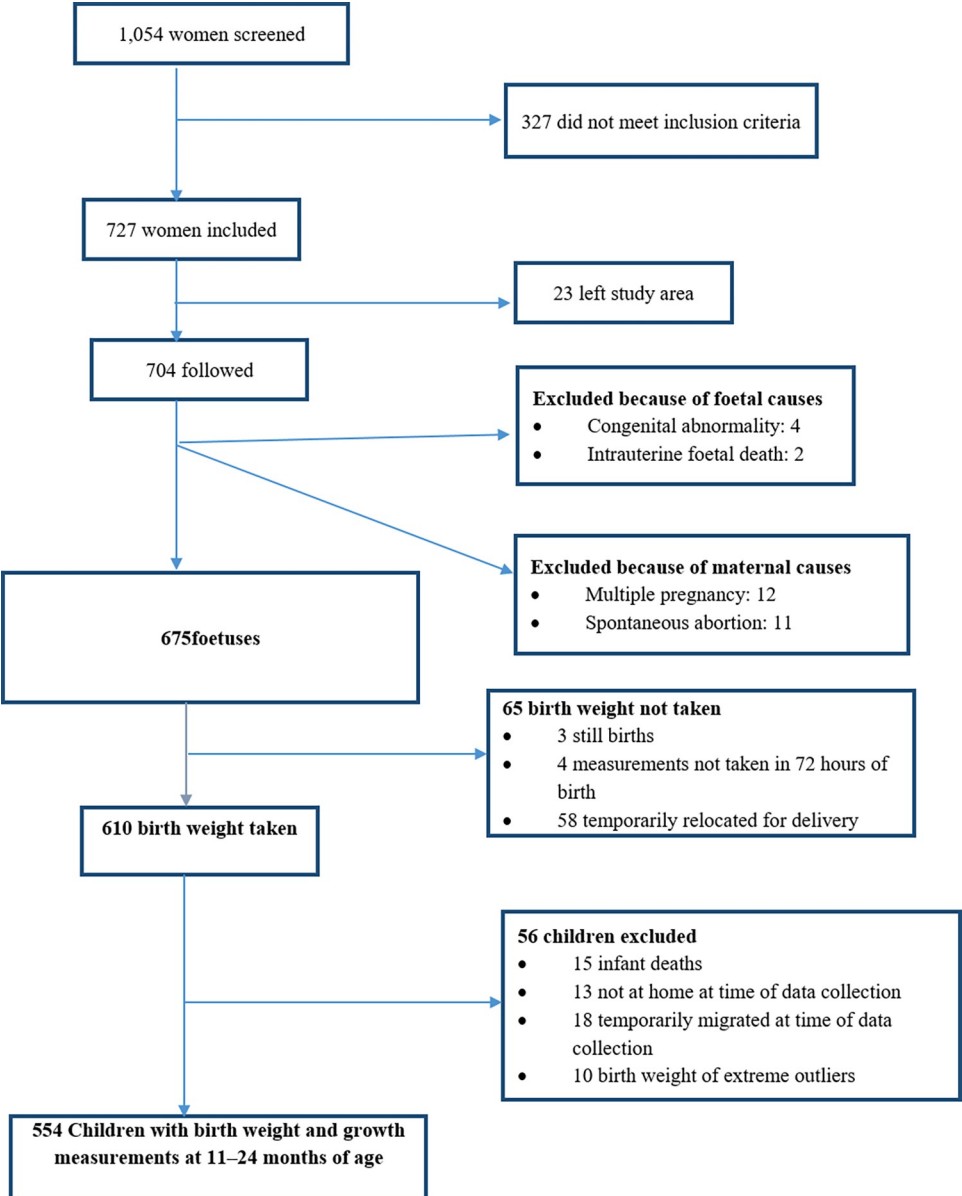

**Fig 2. Flow chart of study participants from fetal to postnatal period at age of 11–24 months, Adami Tullu district (2018).**

After birth, we took weights and lengths of 554 children at age of 11–24 months. Forty-six children did not have weight and length measurements. There were 15 infant deaths reported. Thirteen children were not at home at the time of data collection. Eighteen children migrated out of the study area at time of data collection. Furthermore, 10 children were excluded from the analysis because their recorded birth weight fell in the extreme outlier range (Fig 2).

The households and maternal characteristics of the study participants are shown in Table 1. The mean family size of the study participants was 5.6 individuals per household. Two hundred eight nine (52.7%) of the mothers did not have formal education. Among the mothers, 267 (48.3%) were 15–24 years old and 248 (44.8%) were 25–34 years old. The mean parity was 2.8, and 350 (63.2%) had a history of two or more births. Most of them (523, 94.4%) were housewives.

**Table 1. Household and maternal characteristics of the pregnant mothers at first measurement Adami Tullu district, 2018.**

| Variables | n | % | Median | Inter quartile range (IQR) |
|---|---|---|---|---|
| **Household Characteristics** | | | | |
| Family size | 554 | | 5.6 | 2.3 |
| Household wealth | 550 | | | 333.5 |
| Poor | 175 | 31.8 | | |
| Middle | 185 | 33.6 | | |
| Rich | 190 | 34.6 | | |
| **Maternal Characteristics** | | | Mean | Standard Deviation |
| Height | 547 | | 157.3 | 6.4 |
| Age in years (n = 553) | | | 25.0 | 5.4 |
| 15–24 | 267 | 48.3 | | |
| 25–34 | 248 | 44.8 | | |
| ≥ 35 | 38 | 6.9 | | |
| | | | Median | Interquartile range (IQR) |
| Gravida (n = 554) | | | 4.0 | 4 |
| 1 | 82 | 14.8 | | |
| 2–5 | 307 | 55.4 | | |
| ≥6 | 165 | 29.8 | | |
| Parity (n = 554) | | | 2. | 4 |
| 0 | 95 | 17.1 | | |
| 1 | 109 | 19.7 | | |
| 2–5 | 266 | 48.0 | | |
| ≥6 | 84 | 15.2 | | |
| Occupation (n = 554) | | | | |
| Housewife | 523 | 94.4 | | |
| Others | 31 | 5.6 | | |
| Education (n = 548) | | | | |
| No formal education | 289 | 52.7 | | |
| Primary | 145 | 26.5 | | |
| Junior Secondary | 67 | 12.2 | | |
| Secondary | 47 | 8.6 | | |

We observed a larger family size for mothers who had no formal education (179 (61.9%)) and a lower family size for mothers who had secondary education (12 (25.5%)). Family size was higher for mothers who are housewives (268 (51.2%)) than those who had other occupations (9 (29.0%)). Among the mothers who had no formal education, 95.5% were housewives and 4.5% had other occupations. While among mothers who had secondary education, 85.1% were housewives and 14.9% had other occupation.

## Characteristics of children during pregnancy, birth, and postnatal follow-ups

Table 2 shows characteristics of the cohort of 554 children during fetal, neonatal, and 11–24 months of follow-ups after birth. Overall, we obtained 1,479 ultrasound prenatal biometric measurements. We took 515 ultrasound measurements during 24–28 weeks, 501 ultrasound measurements during 29–33 weeks and 463 ultrasound measurements during 34–38 weeks of pregnancy.

Table 3 shows the characteristics of the cohort of children during neonatal and postnatal follow-ups. The mean birth weight of the new-born was 3,240.7gm (SD 481.0), and 324

**Table 2. Fetal characteristics during pregnancy, Adami Tullu district, 2018.**

| Variable | n | Mean | Standard Deviation |
|---|---|---|---|
| Biparietal Diameter (mm) (n = 1,479*) | | 77.2 | 9.7 |
| 24–28 weeks | 515 | 66.5 | 3.4 |
| 29–33 weeks | 501 | 77.6 | 3.9 |
| 34–38 weeks | 463 | 88.6 | 3.7 |
| Head Circumference (mm) (n = 1,479*) | | 286.5 | 31.8 |
| 24–28 weeks | 515 | 250.3 | 10.7 |
| 29–33 weeks | 501 | 288.8 | 11.5 |
| 34–38 weeks | 463 | 324.2 | 8.1 |
| Abdominal Circumference (mm) (n = 1,479*) | | 259.4 | 40.7 |
| 24–28 weeks | 515 | 215.0 | 12.2 |
| 29–33 weeks | 501 | 258.6 | 15.6 |
| 34–38 weeks | 463 | 309.4 | 13.1 |
| Femoral Length (mm) (n = 1,479*) | | 59.2 | 8.8 |
| 24–28 weeks | 515 | 49.6 | 2.7 |
| 29–33 weeks | 501 | 59.1 | 3.1 |
| 34–38 weeks | 463 | 69.9 | 2.8 |

n: number, mm: millimeter

*: number of biometric measurements.

(58.5%) of them were male. Among neonates, 35 (6%) had low birthweight, and 29 (5.2%) were born before 37 weeks of gestation.

Growth measurements of height and weight were taken for 554 children at 11–24 months of age. The mean age of the children was 17.2 (SD 3.3) months. Among the 554 children, 301

**Table 3. Child characteristics at birth and 11–24 months after birth, Adami Tullu district, 2018.**

| Variable | n | % | Mean | Standard Deviation |
|---|---|---|---|---|
| Birthweight (n = 554) | | | 3,240.7 | 481.01 |
| < 2,500 gm | 35 | 6.3 | | |
| ≥ 2,500 gm | 519 | 93.7 | | |
| Gestational age at birth (n = 553) | | | | |
| < 37 weeks | 29 | 5.2 | | |
| ≥ 37 weeks | 524 | 94.8 | | |
| Sex (n = 554) | | | | |
| Male | 324 | 58.5 | | |
| Female | 230 | 41.5 | | |
| Post-natal age in months (n = 554) | | | 17.2 | 3.3 |
| 11–16 | 231 | 41.7 | | |
| >16–20 | 196 | 35.4 | | |
| >20–24 | 127 | 22.9 | | |
| Length-for-age (n = 554) | | | -2.092 | 1.521 |
| ≤ -2 z-score | 301 | 54.3 | | |
| > -2 z-score | 253 | 45.7 | | |
| Weight-for-length (n = 554) | | | 0.342 | 1.742 |
| ≤ -2 z-score | 59 | 10.6 | | |
| > -2 z-score | 495 | 89.4 | | |

n: number, gm: gram

(54.3%) were stunted, and 59 (10.6%) had wasting. The prevalence of wasting was lower in the age group greater than 20 months (8.7%) as compared to children 11–24 months of age (11.3%). The difference was not statistically significant (95% CI, -0.37, 0.32, P-value = 0.885).

However, stunting was higher (64.6%) among children 20 and above months of age than among children 11–20 months (%). The difference was statistically significant (95%CI, 0.20, 0.79 P-value = 0.001).

## Factors associated with birthweight

Associations between birth weight and measures of fetal ultrasound growth (HC, BPD, AC and FL) during the different times throughout pregnancy are shown in Table 4. The BPD ultrasound measurement at 24–28 weeks [β = 23.41 95% CI: (0.60, 46.22) P-value = 0.444], HC ultrasound measurement [β = 9.72 95% CI: (0.82, 18.63) P-value = 0.032], and abdominal circumference [β = 7.82 95% CI: (3.72, 11.91) P-value = 0.001] at the 34–38 weeks were significantly associated with birth weight. Similarly, an increase in parity [β = 34.31 95% CI: (9.99, 58.63) P-value = 0.006], educational status [β = 17.34 95% CI: (4.28, 30.40) P-value = 0.009] and gestational age at birth [β = 65.58 95% CI: (41.45, 89.72) P-value = 0.001] were associated with increased birth weight.

For birth weight, a mixed-effects linear model was fitted and the Intra cluster correlation (ICC) was estimated as the ratio of the between-kebele variance to the total variance that

**Table 4. Multilevel, mixed-effects regression coefficients for associations of household, maternal, and fetal characteristics with birthweight of children in Adami Tullu district, 2018.**

| Variables | 24–28 weeks | | 29–33 weeks | | 34–38 weeks | |
|---|---|---|---|---|---|---|
| | β (95% CI) | P-Value | β (95% CI) | P-Value | β (95% CI) | P-Value |
| Fixed effects | | | | | | |
| Family size (number) | -3.81 (-22.73, 15.11) | 0.693 | 7.33 (-13.18, 27.84) | 0.483 | -0.09 (-21.09, 20.91) | 0.993 |
| Household wealth status (index) | -0. 05 (-0.26, 0.12) | 0.620 | -0.06 (-0.28, 0.15) | 0.564 | 0.0007 (-0.22, 0.24) | 0.951 |
| Maternal age (year) | -263 (-12.31, 7.052) | 0.595 | -1.77 (-11.73, 8.18) | 0.727 | 2.66 (-7.55, 12.87) | 0.610 |
| Maternal education (grade) | 17.34 (4.28, 30.40) | 0.009 | 13.39 (-0.64,27.42) | 0.061 | 15.30 (0.91,29.68) | 0.037 |
| Maternal height (cm) | 2.93 (-2.22, 11.53) | 0.350 | 2.96 (-3.72, 9.64) | 0.385 | 0.17 (-6.53, 6.89) | 0.959 |
| Parity (number) | 34.31 (9.99, 58.63) | 0.006 | 17.43 (-8.38,43.24) | 0.186 | 21.77 (-5.18, 48.71) | 0.113 |
| Maternal occupation other than housewife (type) | -20.23 (-198.10 157.63) | 0.824 | -134.29 (-330.89, 2.31) | 0.181 | -10.54–47.68 (-232.99, 137.63) | 0.614 |
| Head circumference (mm) | -6.95 (-14.99, 1.10) | 0.091 | -315 (-10.60, 4.31) | 0.408 | 9.72 (0.82,18.63) | 0.032 |
| Biparietal diameter (mm) | 23.41(0.60, 46.22) | 0.044 | 9.69 (-10.54, 29.93) | 0.348 | 16.99 (-35.20, 1.21) | 0.067 |
| Abdominal circumference (mm) | 2.89 (-1.28,7.07) | 0.175 | 5.79(2.05, 9.54) | 0.002 | 7.82 (3.72, 11.91) | 0.001 |
| Femoral Length (mm) | 6.67 (-12.15, 25.49) | 0.488 | -12.36 (-30.62, 5.90) | 0.185 | -0.73 (-18.38, 16.92) | 0.935 |
| Gestational age at birth (week) | 65.58 (41.45, 89.72) | 0.001 | 61.48 (36.003, 86.951) | 0.001 | 32.93 (4.36, 61.50) | 0.024 |
| Sex | -46.90 (-129.06, 35.26) | 0.263 | -33.69 (-118.92, 51.54) | 0.439 | -36.49 (-125.95, 52.98) | 0.424 |
| Random-effects parameters | | SE | | SE | | SE |
| Kebele (constant)* | 18117.5 (5988.5, 54811.7) | 10233.11 | 21914.57 (6726.98, 71391.37) | 6726.98 | 22694.5 (7471.88, 68930.47) | 12864.03 |

* Variation in birth weight among kebeles (Intra-cluster correlation coefficient: ICC (ρ) = 0.11)

ρ: rho

SE: Standard error

CI: Confidence Interval

β: coefficient

cm: centimeter

mm: millimeter

accounts for the relatedness of birthweight data by comparing the variance within kebeles with the variance between kebeles. The estimated ICC (ρ) for birth weight was estimated to be 0.11.

## Factors associated with linear child growth

Table 5 shows the associations between intrauterine growth as measured by ultrasound (HC, BPD, AC and FL) during the different times throughout pregnancy and linear child growth as measured by length-for-age z-score at 11–24 months of the child age. The linear child growth was significantly associated with femoral length measured by ultrasound during 24–28 weeks [β = 0.06 95% CI: (0.01, 0.12) P-value = 0.016], gestational age at delivery [β = 0.07 95% CI: (0.004, 0.142) P-value = 0.038] and 11–24 months child age after birth [β = -0.09 95% CI: (-0.13, -0.05) P-value = 0.001]. An increase in one millimeter of femoral length for 24–28 weeks was associated with a 0.06 increase in length-for-age z-score. An increase in one week of gestational age was associated with a.07 increase in LAZ. Length-for-age was also significantly associated with child age. An increase of child age by one month was associated with 0.11cm decrease of child LAZ. We also found a significant association between family size and length-for-age z-score [β = 0.09 95% CI: (0.04, 0.15) P-value = 0.001]. Child LAZ increased with increasing family size.

**Table 5. Multilevel, mixed-effects regression coefficients for associations of household, maternal, fetal, and child characteristics with length-for-age z-score among children aged 11–24 months in Adami Tullu district, 2018.**

| Variables | 24–28 weeks | | 29–33 weeks | | 34–38 weeks | |
|---|---|---|---|---|---|---|
| | β (95% CI) | P-Value | β (95% CI) | P-Value | β (95% CI) | P-Value |
| Fixed effects | | | | | | |
| Family size (number) | 0.09 (0.04, 0.15) | 0.001 | 0.09 (0.04, 0.15) | 0.001 | 0.07 (0.01, 0.13) | 0.016 |
| Household wealth status (index) | -0.0002 (-0.0008, 0.0003) | 0.422 | -0.0002 (-0.0008, 0.0004) | 0.517 | -0.0004 (-0.0010, 0.0003) | 0.251 |
| Maternal age (year) | -0.002 (-0.028, 0.025) | 0.908 | -0.01 (-0.04, 0.02) | 0.503 | -0.01 (-0.04, 0.02) | 0.605 |
| Maternal education (grade) | -0.01 (-0.05, 0.03) | 0.612 | 0.001 (-0.04, 0.04) | 0.949 | -0.01 (-0.04, 0 .03) | 0.810 |
| Maternal height (cm) | 0.001 (-0.02, 0.02) | 0.945 | 0.001 (-0.02, 0.02) | 0.945 | 0.01 (-0.01, 0.03) | 0.303 |
| Parity (number) | -0.03 (-0.10, 0.04) | 0.396 | -0.01 (-0.08, 0.06) | 0.761 | -0.01 (-0.08, 0.07) | 0.843 |
| Maternal occupation other than housewife (type) | 0.13 (-0.35, 0.62) | 0.594 | -0.25 (-0.78, 0.28) | 0.351 | 0.03 (-0.47, 0.53) | 0.906 |
| Head circumference (cm) | -0.02 (-0.041, 0.003) | 0.100 | -0.002 (-0.02, 0.02) | 0.840 | -0.01 (-0.03, 0.02) | 0.472 |
| Biparietal Diameter (cm) | -0.01 (-0.07, 0.06) | 0.868 | -0.001 (-0.06, 0.05) | 0.969 | 0.02 (-0.03, 0.07) | 0.482 |
| Abdominal circumference (cm) | 0.002 (-0.009, 0.014) | 0.681 | 0.003 (-0.007, 0.013) | 0.531 | 0.01 (-0.01, 0.02) | 0.407 |
| Femoral length (cm) | 0.06 (0.01, 0.12) | 0.016 | -0.01 (-0.05, 0.04) | 0.842 | 0.02 (-0.02, 0 .07) | 0.339 |
| Gestational age at birth (week) | 0.07 (0.004, 0.142) | 0.038 | 0.04 (-0.03, 0.11) | 0.236 | 0.04 (-0.04, 0.12) | 0.349 |
| Birth weight (gm) | -0.0002 (-0.0004, 0.0001) | 0.179 | -0.0001(-0.0004, 0.0001) | 0.370 | -0. 0001 (-0.0004, 0.0001) | 0.393 |
| Sex | -0.19 (-0.42, 0.03) | 0.096 | -0.013 (-0.36, 0.10) | 0.270 | -0.07 (-0.32, 0.17) | 0.565 |
| Child age (month) | -0.09 (-0.13, -0.05) | 0.001 | -0.10 (-0.14, -0.06) | 0.001 | -0.11 (-0.15, -0.08) | 001 |
| Random-effect parameters | | SE | | SE | | SE |
| Kebele* | 0.634 (0.28, 1.44) | 0.264 | 0. .664 (0.292, 1.508) | 0.278 | 0.612 (0 .268, 1.399) | 0.258 |

* Variation in length-for-age among kebeles (Intra-cluster correlation coefficient: ICC (ρ) = 0.30)

ρ: rho

SE: Standard error

CI: Confidence Interval

β: coefficient

cm: centimeter

mm: millimeter

**Table 6. Multilevel, mixed-effects regression coefficients for association of household, maternal, fetal, and child characteristics with weight-for-length z-score among children aged 11–24 months in Adami Tullu district, 2018.**

| Variables | 24–28 weeks | | 29–33 weeks | | 34–38 weeks | |
|---|---|---|---|---|---|---|
| | β (95% CI) | P-Value | β (95% CI) | P-Value | β (95% CI) | P-Value |
| Fixed effects | | | | | | |
| Family size (number) | -0.10 (-0.16, -0.03) | 0.006 | -0.10(-0.17, -0.03) | 0.005 | -0.07 (-0.145, 0.002) | 0.055 |
| Household wealth status (index) | 0.0005 (-0.0002,0.0013) | 0.171 | 0.0007 (-0.00002,0.00150) | 0.055 | 0.0006 (-0.0002, 0.0014) | 0.113 |
| Maternal age (year) | 0.01 (-0.03,0.04) | 0.773 | 0.004 (-0.03,0.04) | 0.831 | -0.002 (-0.04,0.03) | 0.930 |
| Maternal education (grade) | -0.01 (-0.06, 0.03) | 0.587 | -0.02 (-0.07,0.03) | 0.465 | -0.01 (-0.06,0.04) | 0.657 |
| Maternal height (cm) | 0.03 (0.01, 0.05) | 0.011 | 0.03 (0.003, 0.049) | 0.025 | 0.02 (0.001,0.047) | 0.042 |
| Parity (number) | -0.03 (-0.12, 0.05) | 0.461 | -0.03 (-0.12, 0.06) | 0.547 | -0.02 (-0.12,0.07) | 0.625 |
| Maternal occupation other than housewife (types) | -0.43 (-1.06, 0.21) | 0.189 | -0.37 (-1.05, 0.31) | 0.287 | -0.38(-1.02, 0.26) | 0.246 |
| Gestational age at birth (week) | 0.02 (-0.07, 0.11) | 0.722 | 0.04 (-0.05, 0.13) | 0.427 | 0.02 (-0.08, 0.12) | 0.736 |
| Birth weight (gm) | 0.0003 (-0.0001, 0.0006) | 0.113 | 0.0002 (-0.0001, 0.0005) | 0.297 | 0.0002 (-0.0001, 0.0005) | 0.208 |
| Sex | -0.07 (-0.36, 0.23) | 0.651 | -0.07 (-0.37, 0.22) | 0.628 | -0.12 (-0.43, 0.19) | 0.464 |
| Child age (month) | -0.01 (-0.05,0.04) | 0.821 | 0.03 (-0.02, 0.08) | 0.254 | 0.03 (-0.02,0.08) | 0.294 |
| Random-effects parameters | | SE | | | | |
| Kebele* | 0.697 (0.29, 1.66) | 0.309 | 0.793 (0.337, 1.870) | 0.347 | 0.649 (0.270, 1.558) | 0.290 |

* Variation in weight-for-height among kebeles (Intra-cluster correlation coefficient: ICC (ρ = 0.22)

ρ: rho

SE: Standard error

CI: Confidence Interval

β: coefficient

cm: centimeter

gm: gram

For length-for-age z-score, a mixed-effects linear model was fitted and the Intra cluster correlation (ICC) was estimated as the ratio of the between-kebele variance to the total variance. The estimated ICC (ρ) for length-for-age z-score was 0.30.

## Factors associated with weight-for-length Z-score

Weight-for-Length Z-score was associated with family size and maternal height in multivariate multilevel, mixed-effect linear regression analyses. Children from smaller family size had better WLZ [β = -0.01 95% CI: (-0.16, -0.03) P-value = 0.006] (Table 6). Similarly, an increase in maternal height had significant association with WLZ [β = 0.03 95% CI: (0.01, 0.05) P-value = 0.011]. Nonetheless, in our study WLZ was not associated with intrauterine fetal growth factors.

## Discussion

In this prospective cohort study, we found that fetal factors, duration of pregnancy, child age, maternal height and family size were the main predictors of linear child growth measured by LAZ and WLZ in children aged 11–24 months. Both birth weight and LAZ were influenced by early intrauterine fetal growth. Fetal growth during late pregnancy influenced birth-weight only. However, in our study intrauterine fetal growth did not show any influence on WLZ. Moreover, we observed a large variation in LAZ and WLZ between villages (kebeles).The influence of environmental factors is more on the child linear growth than the birth weight. Our study also found that fetal, pregnancy, and maternal factors were significantly associated with birthweight z-score.

Some of the predictor factors (the fetal factor, gestational age at birth and child age) for LAZ that we identified in our study were non-modifiable factors. However, the association with family size, maternal education and area of residence emphasize the importance of modifiable household factor and long-lasting consequences for children's growth. The risk factors we found to have associations with linear growth faltering in our study were also identified in previous studies. The finding of an association between an increase in femoral length with increasing LAZ in our study is similar to the finding from a prospective cohort study that showed a highly significant correlation of femoral length with child growth (LAZ) [31]. An increase in LAZ with increasing gestational age at birth in our study is also similar to a study from prospective cohorts of young children in Ghana, Malawi and Burkina Faso that found gestational age at birth to be strongly associated with child linear growth [13].

The other predictor variable that we found to affect LAZ was child age. LAZ decreased with an increase in child age, which is similar to the finding of a study from 125 demographic health survey conducted in 57 countries that reported a decline of LAZ with increasing child age from birth until 21 months [32]. This can be explained by the fact that growth in early life is rapid and it decreases as the child ages [33, 34]. The other finding in our study is the association of family size with the linear growth of the children. In children from larger family size, the LAZ was better while the WLZ was lower. The finding of lower WLZ in larger family size was also reported in a previous study from Bangladesh [35]. A possible explanation may be that inadequate dietary intake among children, because of larger family size, increases the risk of wasting that tends to peak in early childhood, at the age of about 12 months [36, 37]. However, the evidences from previous studies on the association between family size and LAZ were not conclusive. Some have reported that child growth faltering in LAZ is less likely in those from smaller family size [36, 38, 39]. While another study, that is consistent with the finding of our study, reported that child growth faltering in LAZ is less likely in those from larger family size [40]. Wasting was also associated with maternal height in the current study where those children of taller mothers were less likely to have wasting. This is in line with evidence reported from the previous studies that showed maternal height as one of the risk factors identified for impaired growth in children [35, 41].

Some of the factors associated with birth weight in our study were similar to findings of other studies. Studies conducted in Indonesia showed that maternal education, gestational age at birth, and sex were found to be determinants of birthweight [42, 43]. Similarly studies from India [44] and Nepal [45] found significant associations between gestational age at birth and birthweight. Our study also found that biparietal diameter at early pregnancy and head and abdominal circumference at late pregnancy were significantly associated with birthweight.

The strength of our study was that we enrolled our cohort of women during pregnancy and followed their fetuses and children prospectively for 11–24 months postpartum. This allowed us to examine risk factors throughout pregnancy and early childhood. The study has demonstrated a temporal relationship of the exposure and outcome. The prospective design allowed to draw associations between exposure variables that we measured at earlier time during pregnancy or at birth and those measured later during early childhood, the outcome. It has shown the influence of intrauterine growth during pregnancy or at birth on the child linear growth. The study has also demonstrated a dose-response relationship. For instance, an increase in femoral length was associated with increasing LAZ. Measurement of serial biometry using ultrasound by the primary author that was done to characterize fetal growth could minimize inter observer reliability bias.

This study has some limitations. Firstly, we did not measure factors such as dietary, infections, access to healthcare, or environmental factors that may affect LAZ. Secondly, we did not collect data on some of the risk factors of wasting including infection, micronutrient

deficiencies, and child feeding practices, community sanitation, and access to healthcare. Thirdly, we found a higher male to female proportion in our study compared to a previous study in the same area [22]. The male dominance in our study is an unexpected finding and this introduces selection bias. However, this may not have affected our analysis on linear growth as we have adjusted for sex distribution in the multivariate analysis. Fourthly, the final sample size used in this study was less than the calculated minimum sample size at the beginning of the study because of exclusion at different stages that may have resulted in non-significance association with the confounding variables. We calculated statistical power for the variables and all of them except wealth status had low power. This may indicate that the lack of association might be due to low statistical power of the variables.

In conclusion, our study shows that fetal growth, duration of pregnancy, child age, maternal height, and family size influence linear growth of children. Environmental factors that are associated with the area of residence play a bigger role for linear growth than for birth weight.

## Acknowledgments

Our gratitude goes to all the women and their children who participated in this study. We would also like to thank the data collectors and supervisors involved in this study. We are grateful to the Obstetricians who provided training of ultrasound and validated the ultrasound measurements. Our appreciation also goes to Addis Ababa University and the University of Bergen for their support.

## Author Contributions

**Conceptualization:** Meselech Roro, Wakgari Deressa, Bernt Lindtjørn.

**Data curation:** Meselech Roro, Wakgari Deressa, Bernt Lindtjørn.

**Formal analysis:** Meselech Roro, Wakgari Deressa, Bernt Lindtjørn.

**Funding acquisition:** Wakgari Deressa, Bernt Lindtjørn.

**Investigation:** Meselech Roro.

**Methodology:** Meselech Roro, Wakgari Deressa, Bernt Lindtjørn.

**Project administration:** Meselech Roro, Wakgari Deressa, Bernt Lindtjørn.

**Resources:** Wakgari Deressa, Bernt Lindtjørn.

**Software:** Meselech Roro, Wakgari Deressa, Bernt Lindtjørn.

**Supervision:** Meselech Roro, Wakgari Deressa, Bernt Lindtjørn.

**Validation:** Meselech Roro, Wakgari Deressa, Bernt Lindtjørn.

**Visualization:** Meselech Roro.

**Writing – original draft:** Meselech Roro, Wakgari Deressa, Bernt Lindtjørn.

**Writing – review & editing:** Meselech Roro, Wakgari Deressa, Bernt Lindtjørn.

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
