## [Decision Letter · Decision Letter 0]

23 May 2022

PONE-D-21-30564Influence of intrauterine factors on birth weight and on child linear growth in rural Ethiopia:  A prospective cohort studyPLOS ONE

Dear Dr. Roro,

Thank you for submitting your manuscript to PLOS ONE. After careful consideration, we feel that it has merit but does not fully meet PLOS ONE’s publication criteria as it currently stands. Therefore, we invite you to submit a revised version of the manuscript that addresses the points raised during the review process.

Your manuscript has been assessed by two reviewers, whose comments are appended below. Both reviewers raise important concerns about several aspects of the methodology and presentation, which you should address carefully in your revised manuscript.

We look forward to receiving your revised manuscript.

Kind regards,

Joseph Donlan

Editorial Office

PLOS ONE

Journal Requirements:

4. Please upload a new copy of Figure 1 as the detail is not clear. Please follow the link for more information: https://blogs.plos.org/plos/2019/06/looking-good-tips-for-creating-your-plos-figures-graphics/" https://blogs.plos.org/plos/2019/06/looking-good-tips-for-creating-your-plos-figures-graphics/

5. Please upload a copy of Figure 3, to which you refer in your text on page 11. If the figure is no longer to be included as part of the submission please remove all reference to it within the text.

Reviewers' comments:

Reviewer's Responses to Questions

**Comments to the Author**

1. Is the manuscript technically sound, and do the data support the conclusions?

Reviewer #1: Yes

Reviewer #2: Yes

2. Has the statistical analysis been performed appropriately and rigorously? 

Reviewer #1: Yes

Reviewer #2: No

3. Have the authors made all data underlying the findings in their manuscript fully available?

Reviewer #1: Yes

Reviewer #2: Yes

4. Is the manuscript presented in an intelligible fashion and written in standard English?

Reviewer #1: Yes

Reviewer #2: Yes

5. Review Comments to the Author

Reviewer #1: The title of the manuscript clearly expresses what the study is about and highlights the importance of the study.

Although the abstract presents the aim, methods, important findings, and conclusion, it appears too long as the word count exceeds the specified word limit. As such, summarizing the sub-sections will improve the abstract.

Data analysis is appropriate. However, the measure of central tendency ('mean') used to summarize the following variables gravidity, family size, and parity, is inappropriate as they are not continuous variables. Suggestion: Mode.

The methods are appropriate for the research questions but the final sample size after exclusion at different stages is less than the calculated minimum sample size. See lines 259-268 and 275. 'increased' non-response rate during sample size calculation would have been helpful in dealing with the above challenge.

-Kindly limit the use of first-person pronouns in the report ( e.g see Lines 91,97 and 106 in the manuscript). Rephrasing such sentences may improve the manuscript.

Overall, it is very good work.

Reviewer #2: Review and comment of the manuscript

Title: Influence of intrauterine factors on birth weight and on child linear growth in rural Ethiopia: A prospective cohort study. PGPH-D-21-30564

Comments to the author

General comments:

• The paper is well designed but the manuscript sections have to be consistent and some edit need to attract attention from the authors. Method section should need more details to define variables and to explain the statistically data analysis.

• The authors should pay more attention to the Results section and subsequently Discussion section.

• Format and consistency challenges need few restructuration. However, the author should pay more attention to be consistent with definitions and spelling of key variables.

• The effect size estimates, their values and the significance level are missing in the text and tables.

Specific comments:

Introduction

• Method; study design section:

o Lines 106-108: avoid repetitions in the sentence that doesn’t add value. “…that influence growth…. That affect linear growth…”

o Lines 110 to 112: this sentence can be restructured in one sentence. “…foetal and child factors including foetal factors (HC, BP…) and birth characteristics (age, sex) …”

o Anthropometric measurements: Authors should provide how the quality control and precision have been done for maternal and child anthropometric measurements.

o Nothing is said about how the wealth status has been computed. Which assets have been included for principal component analyses?

Statistical analysis

o What was the added value to use two software (SPSS and Stata) for the same results, the same study? Using two software, how the authors addressed the conflicting statistical results between them?

o Lines 212-214: the author needs to add the third undernutrition status which is underweight related to weight-for-height z-score index.

o Lines 233-242: do gender and sex variables have the same meaning in the text?

• Participants section:

o Line 126 should be edited: “…have been …”

o The author has to state based on the evidence with references and not from news in the first line of the paragraph.

o References must be inserted for the follows section of the paragraph: “….were postponed.”

• Paragraph 3: the author must avoid repetitions in the text such the aim in lines 5 and 10. And please, restructure the sentence: “…to answer these questions, this study aims to understand the experiences…”

• The author has to remove bold and bullets from the specific objectives section. To understand the level of the stress, I doubt that the author can use a quantitative analysis as mentioned in the following method section. The author can use another verb.

Methods

• Setting section (general setting and specific setting) has to be merged, shortened, and more accurate. Some other information is not relevant to the study aims and the study setting section such as the human index. The meaning of “lakh” seems confused and has to be reworded. The author has to delete the section that does not add any value:” …and is still in place with relaxations in the state.”

• Study population:

o Eligibility criteria according to age must be inserted in this section. The interview was either semi-structured (method section) or structured (abstract). Please, clarify the interview type?

• Variables, data source and data collection

o Definitions and the meaning and categories of all variables (exposures and outcomes) are missing. It is clear shown the unit for variables instead of the clear definitions of them. Authors should define variables clearly. For example, writing like “family size (number)” is confusing and not sufficient either defining family size or precising the unit of family size data.

o In the method section, authors should clarify how data have been collected, aggregated and analysed to express Household wealth through principal component analysis method.

o The collectors of data are not mentioned in the study, and which measurements they used to collect data and each variables. Authors should provide how they develop the quality control for measurement for their accuracy and reproducibility during the study.

o The author must avoid repetitions in the whole text. Repetition in line 4 and 12 in the first paragraph for variables and data sources through the data collection section.

• The analysis must state the effects size estimates that have been used to express the results in the statistical analyses section (Multilevel, mixed effect regression). Did this statistical analysis fit with data and the study design (cohort study)? The confidence interval and level of significance are missing from the methods and results sections.

Results

• Results are relevant but text section is not linked to the five tables. How has the author proceeded to the regression analyses? Nothing has been said about the binary and multivariable steps. How the author managed the confusion factors from the regression models?

• Tables need to have the legend, confidence intervals, and symbols need to be explained.

• Table 1 has some formats that need to be edited. What is the meaning of mean for household wealth? The summary of household wealth variable should consider poor and rich households instead of the mean in table 1.

• Table 2 and 3: while we assume that twins and multiple pregnancy were excluded from the study, the authors should clarify disparate samples with more children compared to mothers. For instance, 1479 children in table 2 compared to 547 women in table 1. Otherwise, the sample “n” does confuse readers when we consider the study design which is a cohort study with loss to follow up participants from the enrolment to the last time in the pregnancy. We can’t compute the mean for the sum of number of the participants that we follow at different time-points.

• Table 4: clearly, maternal education and child age at birth remain the key factor for birth weight (strong relationship). Child anthropometry is associated to birth weight too. This statement should be developed in the discussion.

• Table 5: family size and child age are associated to LAZ and WAZ. The authors should provide the direction for this relationship.

• The first paragraph should insert the finding from tables 5 and 6 as stated in this review. Was there collinearity between maternal education and child age at birth or birth weight?

References

• The use of capital letter in the titles (either for the first letter of the title or first letter for each word) should be consistent for the whole list of references.

• Check for the use of capital letters in titles, abbreviation for the journal, the consistency of pages. Some references don’t have number of titles.

• Reference 19 must be complete.

6. PLOS authors have the option to publish the peer review history of their article (what does this mean?). If published, this will include your full peer review and any attached files.

Reviewer #1: **Yes: **Dr. Felix C.C. WEKERE

Reviewer #2: **Yes: **Christian Bwangandu Ngandu

---

## [Author Response · Author response to Decision Letter 0]

11 Jul 2022

We have attached our point by point response to comments from the editor and reviewers : File name " response to reviewers"

---

## [Editor Report · Decision Letter 1]

25 Jul 2022

Influence of intrauterine factors on birth weight and on child linear growth in rural Ethiopia:  A prospective cohort study

PONE-D-21-30564R1

Dear Dr. Roro,,

We’re pleased to inform you that your manuscript has been judged scientifically suitable for publication and will be formally accepted for publication once it meets all outstanding technical requirements.

Kind regards,

Felix C.C. Wekere

Guest Editor

PLOS ONE

Additional Editor Comments (optional):

Thank you for revising the manuscript titled “Influence of intrauterine factors on birth weight and on child linear growth in rural Ethiopia: A prospective Cohort Study” (PONE-D-2130564R1), according to PLOS ONE requirements and also addressing the issues raised by the Editor and Reviewers. Having reviewed the manuscript following your initial submission to the Journal, I can confirm that the revised version is presented in an intelligible fashion, easy to read, and well written in standard English. All the review comments have been addressed satisfactorily. I, therefore, recommend its acceptance for publication in PLOS ONE.

However, the following edits may be helpful in the final version to be published:

Abstract

line 24 - 'week' to 'weeks'

line 25- 'centralEthiopia' to 'central Ethiopia'

line 30- 'new-born' to 'newborn

line 32- 'Prevalence' to read 'The prevalence'

line 36- 'significant' to read ' significantly'

Introduction

line 47- ..Phase.. to read 'Phases'
---

## [Editor Report · Acceptance letter]

28 Jul 2022

PONE-D-21-30564R1 

Influence of intrauterine factors on birth weight and on child linear growth in rural Ethiopia:  a prospective cohort study 

Dear Dr. Roro:

I'm pleased to inform you that your manuscript has been deemed suitable for publication in PLOS ONE. Congratulations! Your manuscript is now with our production department. 

Kind regards, 

on behalf of

Dr. Felix C.C. Wekere 

Guest Editor

PLOS ONE